# Plasmon Damping Rates in Coulomb-Coupled 2D Layers in a Heterostructure

**DOI:** 10.3390/ma15227964

**Published:** 2022-11-11

**Authors:** Dipendra Dahal, Godfrey Gumbs, Andrii Iurov, Chin-Sen Ting

**Affiliations:** 1Texas Center for Superconductivity and Department of Physics, University of Houston, Houston, TX 77204, USA; 2Department of Physics and Astronomy, Hunter College, City University of New York, 695 Park Avenue, New York, NY 10065, USA; 3Department of Physics and Computer Science, Medgar Evers College, City University of New York, Brooklyn, NY 11225, USA

**Keywords:** plasmon, graphene, silicene, heterostructure

## Abstract

The Coulomb excitations of charge density oscillation are calculated for a double-layer heterostructure. Specifically, we consider two-dimensional (2D) layers of silicene and graphene on a substrate. From the obtained surface response function, we calculated the plasmon dispersion relations, which demonstrate how the Coulomb interaction renormalizes the plasmon frequencies. Most importantly, we have conducted a thorough investigation of how the decay rates of the plasmons in these heterostructures are affected by the Coulomb coupling between different types of two-dimensional materials whose separations could be varied. A novel effect of nullification of the silicene band gap is noticed when graphene is introduced into the system. To utilize these effects for experimental and industrial purposes, graphical results for the different parameters are presented.

## 1. Introduction

A huge number of researchers from various disciplines have been showing their interest in new materials, silicene especially, after the development of its fabrication process in 2012 [1]. Because of its exceptional potential applications in electronic and optoelectronic devices, many industries are making substantial investments to harness its properties. Additionally, before making investments for commercial gain, both theoreticians and experimentalists have been exploring this material for many years. A credit of foremost importance goes to Takeda and Shiraishi [2], who, in 1994, dealt with the atomic and electronic structure of the material for the first time. These authors calculated the band structure of silicon in the corrugated stage having optimized atomic geometry. This work, though very novel, did not receive the attention it deserves until 2004, when single-layer carbon atoms named graphene were fabricated in the laboratory from graphite by Novoselov et al. [3]. Their research not only validated the stability of two-dimensional (2D) material but also opened the door for new research on thin film materials, silicene being one of them.

Both silicene and graphene were studied in parallel. The former has a buckled crystal geometry, whereas the latter has a honeycomb planar geometry. Due to this, differences arise between them. Ab initio calculations showed that the bandgap of silicene is electrically tunable [4,5,6], which is an advantageous property for designing a field effect transistor that works at room temperature. Another distinct difference between these two materials is the strength of the spin-orbital coupling (SOC), which is very weak in graphene. Consequently, the quantum spin Hall effect occurs at extremely low temperatures [7,8]. In contrast to this, silicene displays quantum spin Hall effect at temperature 18 K, far higher than that for graphene.

Several investigations have been carried out on both graphene and silicene with respect to transport phenomena [9,10,11,12,13,14,15,16], as well as their magnetic and electric field effects [3,17,18,19,20,21,22,23], the fabrication process [24,25,26,27], plasmonic behavior [28,29,30,31,32,33,34,35,36,37,38] and the doping effect on layered graphene and graphene-like heterostructure systems [39,40].

Recently, Dong et al. [41] studied the plasmonic behavior and its decay mode in multilayer graphene structure, however, an extensive literature search on the plasmon-related studies indicates that there has been no investigation regarding plasmon excitations and their decay rates due to Landau damping for composite silicene and graphene materials. This hybrid material could have significant benefits for use in the advancement of quantum information technology [42,43,44], sensing devices [45,46,47] and protein analytic clinical devices, [47,48,49,50]. Based on these impressive potential applications, we are motivated, in this work, to choose a system composed of silicene and graphene accompanied by a conducting substrate. A detailed review of the plasmon properties in graphene and various graphene-based structures has been presented in Ref. [51].

The plasmon mode is tunable by the thickness of the substrate and the variation of material behavior. We first determine the surface response function of the structure, the same technique used recently by Gumbs et al. [52,53], which gives us the condition for the existence of the plasmon dispersion. The analytical result for the surface response function is further used for different limiting cases and a comprehensive comparison is made with a variety of structures composed of different graphene–silicene compositions. Furthermore, the same function is used to obtain the Landau damping rate of the plasmon modes whose numerical calculation demonstrates that its variation depends on the layer separation, types of dielectric used and the type of 2D layer employed.

We have organized the rest of our paper as follows. In Section 2, we present the core idea of our work where we show the analytical result for the surface response function for the chosen structure. Under limiting conditions, the result is used to derive the results for a variety of conditions. The graphical results and their interpretation are presented in Section 3. We conclude our paper with a summary of our main results and conclusions in Section 4.

## 2. Theory

In this section, we analyze the plasmonic behavior of the heterostructure consisting of graphene and silicene together for which we employ the low-energy form of the Hamiltonian near the K point in the Brillouin zone. One significant difference between the Hamiltonian of graphene and silicene is that a small band gap, Δ is present in the silicene energy band structure, which is due to spin-orbit coupling and applied external electric field. This band gap is not seen in intrinsic graphene.

### 2.1. Silicene

We now briefly describe the case pertaining to silicene whose Hamiltonian in the continuum limit is given by:(1)$$H_{\xi }=\hbar v_{F}\left(\xi k_{x}{\hat{\tau }}_{x}+k_{y}{\hat{\tau }}_{y}\right)-\xi \Delta_{so}\sigma_{z}\tau_{z}+\Delta_{z}{\hat{\tau }}_{z}\;,$$
where ${\hat{\tau }}_{x,y,z}$ and $\sigma_{x,y,z}$ are Pauli matrices corresponding to two spin and coordinate sub-spaces, ξ=±1 for the *K* and $K^{\prime }$ valleys, $v_{F}\left(\approx 5\times 10^{5}\right)$ m/s is the Fermi velocity for silicene [5,54], $k_{x}$ and $k_{y}$ are the wave vector components measured relative to the K points. The first term represents the low-energy Hamiltonian, whereas the second term denotes the Kane–Mele system [7] for intrinsic spin-orbit coupling with an associated spin-orbit band gap of $\Delta_{so}$ (of order 5 to 30 meV can could reach up to 100 meV) [55]. The last term in the expression describes the sublattice potential difference that arises from the application of a perpendicular electric field. Equation (Equation 1) for the Hamiltonian is a block diagonal in 2×2 matrices labeled by valley (ξ) and spin σ=±1 for up and down spin, respectively. These matrices are given by [55]:(2)$${\hat{H}}_{\sigma \xi }=\left(\begin{array}{cc} -\sigma \xi \Delta_{so}+\Delta_{z} & \hbar v_{F}\left(\xi k_{x}-ik_{y}\right)\\ \hbar v_{F}\left(\xi k_{x}+ik_{y}\right) & \sigma \xi \Delta_{so}-\Delta_{z} \end{array}\right)\;.$$

This gives the low-energy eigenvalues as:(3)$$E_{k}=\pm \sqrt{\hbar^{2}v_{f}^{2}{\left|k\right|}^{2}+\Delta_{\xi \sigma }^{2}}$$
where $\Delta_{\sigma \xi }=\left|\sigma \xi \Delta_{so}-\Delta_{z}\right|$.

### 2.2. Graphene

The low-energy model Hamiltonian for monolayer graphene is similar to that in Equation (Equation 2) with the diagonal terms replaced by zero and ξ labeling the valley. In this regime, the Hamiltonian for intrinsic graphene is given by [56]:(4)$$\hat{H}=\hbar v_{F}\left(\begin{array}{cc} 0 & \left(\xi k_{x}-ik_{y}\right)\\ \left(\xi k_{x}+ik_{y}\right) & 0 \end{array}\right)$$
with the linear energy dispersion, $E_{k}=\pm \hbar v_{f}\left|k\right|$ in either valley.

### 2.3. Polarization Function: Π(q,ω)

Considerable work has been completed on the dynamical properties involving the use of the dielectric function ϵ(q,ω) of various types of free-standing 2D systems [57,58,59] under different conditions. These include temperature effects [60,61], the role of an ambient magnetic field for the 2D electron gas (2DEG), graphene, silicene [62] and the dice lattice [58]. For a single 2D layer, one can extract the plasmon dispersion relation and damping rate by employing the dielectric function. However, the situation is more complicated for a multi-layer heterostructure that relies on knowledge of the surface response function, which we have presented in detail below. However, in either case, we need to calculate the polarization function obtained in the random phase approximation (RPA). For a 2D layer surrounded by a medium with dielectric constant $\epsilon_{b}$, the dynamic dielectric function is given by:(5)ϵ(q,ω)=1−V(q)Π(q,ω),
where $V\left(q\right)=\frac{2\pi e^{2}}{4\pi \epsilon_{0}\epsilon_{b}q}$ is the Coulomb interaction potential and $\epsilon_{0}$ is the permittivity of free space, *q* is the wave vector and *e* is the electron charge. The polarization function is an important quantity in calculations of the transport, collective charge motion and charge screening properties of the material. In the one-loop approximation, the polarization function for gapped graphene is given by [63]:(6)$$\begin{array}{ccc} \Pi^{0}\left(q,\omega \right) & = & \int \frac{d^{2}k}{2\pi^{2}}\sum_{s,s^{\prime }=\pm 1}\left\{\frac{\hbar^{2}v_{f}^{2}\left(k+q\right)\cdot k+\Delta_{\sigma ,\xi }^{2}}{E_{k}\cdot E_{\left|k+q\right|}}\right\}\frac{f_{0}\left(sE_{k}-E_{F},T\right)-f_{0}\left(s^{\prime }E_{\left|k+q\right|}-E_{F},T\right)}{sE_{k}-s^{\prime }E_{\left|k+q\right|}-\hbar \left(\omega +i\delta \right)}\; \end{array}$$
where the angle between k and k + q is $\theta_{k,k+q}$. At zero temperature, the Fermi function $f_{0}\left(z\right)$ is just a step function. The analytical expression for the polarization function for the silicene and graphene monolayer is given by Tabert et al. [62] and Wunsch et al. [57], respectively.

We now turn our attention to a crucial consideration in this paper regarding the structure consisting of a silicene layer, a graphene layer and substrates as depicted in Figure 1. The dielectric constants $\epsilon_{1}$ and $\epsilon_{2}$ are related to the space between the two layers and the semi-infinite region underneath the lower layer. Thus, we assume that there is no material above the upper layer whose susceptibility is $\chi_{1}\left(q,\omega \right)$, and it is always assumed to be a vacuum above this top layer.

The two 2D layers may be identical or different, possessing different material properties (graphene or silicene in our case), which is reflected in their energy dispersions though the presence or absence of a band gap. The two layers could also have different or equal doping levels (Fermi energies). By employing the boundary condition of continuity of the electrostatic potential and the discontinuity of the electric field across the interface separating two media, we solved for the various coefficients appearing in the potential. Consequently, the result for the surface response function g(q,ω) gives the required conditions for the plasmon dispersion for our case, namely:(7)$$\begin{array}{ccc} & & \phi_{<}\left(z\right)=e^{-qz}-g\left(q,\omega \right)e^{qz}\;,z≲0\;,\\ & & \phi_{>}\left(z\right)=a_{1}e^{-qz}+b_{1}e^{qz}\;,0\leq z\leq d\;,\\ & & \phi_{1>}\left(z\right)=k_{1}e^{-qz}\;,z\geq d\;. \end{array}$$

Here, $\phi_{<}\left(z\right)$, $\phi_{>}\left(z\right)$ and $\phi_{1>}\left(z\right)$ correspond to the electrostatic potential of regions (I),(II) and (III), respectively, as shown in Figure 1. In order to conduct numerical computation, we make use of linear response theory, for which we have the charge density, $\sigma_{1}=\chi_{1}\phi_{<}\left(0\right)$, $\sigma_{2}=\chi_{2}\phi_{1>}$, with $\chi_{1}$, $\chi_{2}$ 2D susceptibilities. Generalizing, $\chi_{i}=e^{2}\Pi_{i}^{0}$ for convenience, we obtain the solution of these equations for different coefficients, leading to:(8)$$g\left(q,\omega \right)=\frac{1}{D(q,\omega )}\left\{\left[q\epsilon_{0}\left(\epsilon_{1}-1\right)-\chi_{1}\right]\left[q\epsilon_{0}\left(\epsilon_{1}+\epsilon_{2}\right)-\chi_{2}\right]-\left[q\epsilon_{0}\left(\epsilon_{1}+1\right)+\chi_{1}\right]\left[q\epsilon_{0}\left(\epsilon_{1}-\epsilon_{2}\right)+\chi_{2}\right]e^{-2dq}\right\}$$
(9)$$D\left(q,\omega \right)\equiv \left[q\epsilon_{0}\left(\epsilon_{1}-1\right)-\chi_{1}\right]\left[q\epsilon_{0}\left(\epsilon_{1}+\epsilon_{2}\right)-\chi_{2}\right]-\left[q\epsilon_{0}\left(\epsilon_{1}-1\right)+\chi_{1}\right]\left[q\epsilon_{0}\left(\epsilon_{1}-\epsilon_{2}\right)+\chi_{2}\right]e^{-2dq}\;,$$
where $\epsilon_{1}\left(\omega \right)$ is the dielectric function of the substrate between layers “1” and “2”, $\chi_{1}$ and $\chi_{2}$ correspond to the susceptibilities of these two layers and *d* is the thickness of the substrate. This substrate thickness is in the order of the wavelength of light considered, for a visible light it could be of the order of a few hundred nanometers. For a thick substrate, the thickness could go up to micrometer in size, and accordingly, the plasmonic mode is modified. The plasmon dispersion equation is obtained from the solutions of D(q,ω)=0, which we solve below. We note that when we set $\chi_{2}=0$ and take the limit d→∞, Equation (Equation 8) gives the well-established form [64]:(10)$$g_{2D}\left(q,\omega \right)=1-{\left\{\frac{\epsilon_{1}+1}{2}-\frac{\chi_{1}}{2q\epsilon_{0}}\right\}}^{-1}\;,$$
which is the surface response function for a 2D layer embedded in a medium whose average background dielectric constant is $\epsilon_{b}=\left(\epsilon_{1}+1\right)/2$. The plasma resonances, which Equation (Equation 10) gives from its poles, are in agreement with the zeros of the dielectric function in Equation (Equation 5).

### 2.4. Damping Rate

We now turn to a critical issue in this paper, which concerns the rate of damping of the plasmon modes by the single-particle excitations. If this rate of damping for a plasmon mode with frequency $\Omega_{p}$ is denoted as γ, then $D(\Omega_{p}+i\gamma ,q)=0$ is the complex frequency space. Carrying out a Taylor series expansion of both the real and imaginary parts, we have:(11)D(Ωp+iγ,q)=ReD(Ωp+iγ,q)+iImD(Ωp+iγ,q)=ReD(Ωp)+iγ∂∂ΩReD(Ω)ω=Ωp+iImD(Ωp)−γ∂∂ΩImD(Ω)ω=Ωp+⋯

Therefore, setting the function in Equation (Equation 11) equal to zero, we obtain γ to the lowest order as:(12)γ=−ImD(Ωp)∂ReD(ω)/∂ω|Ωp.

With these formal results, we now evaluate the plasma spectra for a double layer heterostructure. The expression shows the dependence of γ on the imaginary part of $D(\Omega_{p})$ and the Real part of D(ω), which in turn, are dependent on the type of layer and the substrate considered. Eventually, we can infer that the viability of plasmon modes can be tuned by the dielectric substrate thickness and by the choice of 2D layer. In addition, the rate of decay also helps us in maintaining the intensity and the frequency of the obtained plasmon mode. This could have great impact in the development of the quantum information sharing technology and the data storing devices.

## 3. Numerical Results and Discussion

In our numerical calculations, energy is scaled in units of $E_{F}^{\left(0\right)}$ and the wave vector is scaled with $k_{F}^{\left(0\right)}=\sqrt{\pi n}$, which is in the experimental range for electron/hole doping densities $n=10^{10}$ per cm${}^{2}$. This gives $k_{F}^{\left(0\right)}=10^{6}$ per cm and $E_{F}^{\left(0\right)}$ is equivalent to ∼60 meV. From the preceding discussion, in Section 2, it is clear that the plasmon modes for any system are given by the zeros of the dielectric function obtained from Equation (Equation 9). Thus, making use of this dispersion equation, we computed the plasmon mode dispersion relation for a heterostructure based on graphene and silicene with/without a substrate. For this, we first obtained a graphical result for graphene, as shown in Figure 2. One can clearly see that a single branch plasmon mode originates from the origin in q−ω space, which increases monotonically and is subject to Landau damping when the plasmon mode reaches the interband particle-hole excitation region. Figure 2 is plotted for monolayer graphene embedded in material with background dielectric constant $\epsilon_{1}=\epsilon_{2}$, which we set equal to 1 for simplicity. The situation is similar ($\epsilon_{1}=\epsilon_{2}=1$) for Figure 3, Figure 4, Figure 5, Figure 6 and Figure 7. In Figure 2, the plasmon branches for two values of the Fermi energy are shown in panels (a,b). The damping rates of these plasmon modes are demonstrated in panel (c,d), where it is distinctly shown by an arrow pointing at the boundary of the region where Landau damping takes place. The rate of decay for both types of graphene are monotonically increasing, signifying that the deeper into the single particle excitation region plasmon mode enters, the larger the rate of decay becomes. This implies that the lifetime of the plasmon mode is decreased in the same manner.

Going next to the case when we have a structure with two graphene layers together separated by various distances, a set of plots (as shown in Figure 3) is obtained with two branches of plasmon modes originating from the origin in the q−ω space. One can see that when the graphene sheets are brought closer, the plasmon modes move further apart, signifying weak interactions between the modes. In Figure 3a–d, plasmon modes for a structure with two graphene layers separated by a distance of $0.5{\left(k_{F}^{\left(0\right)}\right)}^{-1}$, $1.0{\left(k_{F}^{\left(0\right)}\right)}^{-1}$, $2.0{\left(k_{F}^{\left(0\right)}\right)}^{-1}$ and $5.0{\left(k_{F}^{\left(0\right)}\right)}^{-1}$ are shown, respectively, and all the plots portray that the further apart the graphene layers are, the closer the two plasmon branches become. For this same set of figures with the other parameters remaining the same, the plasmon decay rate is shown in Figure 4, which shows that the plasmon decay for the lower plasmon branch always starts at a larger wave vector value in comparison to the upper plasmon branch. As the distance of separation is increased, the two plasmon branches come closer and their decay also starts from the same value of the wave vector and the rate of decay is almost the same in value.

Now, in addition, we carried out an investigation of the plasmon modes and their decay rate for the structure with two silicene layers for various separations. The graphical results for these calculations are shown in Figure 5 where we again have two plasmon modes originating from the origin of the q−ω plane. As in the case of a two-graphene-layer structure, we again notice a similar effect on two plasmon branches coming closer to each other when their separation increases. This is demonstrated in Figure 5a–d for the layers separation distance of $0.5{\left(k_{F}^{\left(0\right)}\right)}^{-1}$, $1.0{\left(k_{F}^{\left(0\right)}\right)}^{-1}$, $2.0{\left(k_{F}^{\left(0\right)}\right)}^{-1}$ and $5.0{\left(k_{F}^{\left(0\right)}\right)}^{-1}$, respectively.

As a representative calculation, we investigated the decay rate of plasmon modes for silicene–silicene structure when their separation is $d=0.5{\left(k_{F}^{\left(0\right)}\right)}^{-1}$ and $d=5.0{\left(k_{F}^{\left(0\right)}\right)}^{-1}$. Figure 6 shows that the upper plasmon mode does not decay at all and the lower plasmon branch decays after reaching a critical wave vector. This behavior is due to the presence of a band gap for silicene, resulting in an opening in the single particle excitation region, which provides a larger area in q−ω space for the plasmon mode to survive. The upper plasmon branch in this case has a larger space and is more likely to self-sustain for a longer period without damping. On the other hand, the lower plasmon branch enters the intraband single-particle excitation region where it decays. The rate of decay starts from a critical value of the wave vector and the magnitude of this decay rate monotonically increases.

A comparison of plasmon modes and their decay for graphene–graphene and silicene–silicene structures is shown along with the single-particle excitation regions in Figure 7. The figure in panel (a) of Figure 7 shows that two plasmon modes that stem from the origin of the frequency-momentum space increase steadily and decay when it reaches the boundary of the single-particle excitation region. Corresponding red and blue lines are drawn to further clarify the point where the actual decay begins. The dark triangular region is the area where the plasmon mode survives without Landau damping and mathematically, in this region, the imaginary part of the polarization function of graphene is zero. This means that the plasmon mode has self-sustaining oscillations. The green region where the imaginary part of the polarization function is nonzero is the single particle excitation region where the plasmon mode decays into particle-hole mode. The corresponding decay rate figure, below this panel, shows that the rate of decay for the upper plasmon branch is greater and its critical wave vector is smaller compared to the lower plasmon branch.

Similar plots for silicene–silicene structures were demonstrated in Figure 7b where one can see the opening of a gap in the single-particle excitation region yielding two parts, which is a significant effect arising from the band gap. The imaginary part of the polarization function in this gap region is zero where the plasmon mode can sustain its oscillation for a long time. The upper breakaway region is a single-particle excitation region due to interband transitions of electrons from the valence to the conduction band and the lower breakaway region is the intraband single-particle excitations region, which is due to transitions within the same band from below to above the Fermi level. In Figure 7b, two plasmon modes originate from the origin as demonstrated in the figure. The upper plasmon mode survives without damping over a wide range of wave vectors and the plasmon branch enters the gap created by the opening within the single-particle excitation region. The corresponding decay rate appearing below the plasmon dispersion shows that the upper plasmon branch does not decay at all, whereas the lower plasmon branch with the part in closer contact with the intraband single-particle excitation region has a small plasmon decay rate as illustrated in the corresponding figure in the panel of Figure 7d below. As the plasmon mode rises, it is separated from the single-particle excitation region where the decay rate is zero and as it moves further away from the origin, the plasmon branch becomes closer to the single-particle excitation region where we notice the Landau damping again. Correspondingly, the decay rate increases monotonically and reaches a maximum before dropping down, indicating the reappearance of an undamped plasmon branch at a larger value of the wave vector. Another noticeable effect observed here is the closeness of the plasmon branches and the plasmon decay rate, which can be altered by altering the layer separation; this effect may be used as another plasmon mode tuning parameter.

These two branches appear as a result of the Coulomb interaction between the two layers, which couple the plasmon excitations arising on each layer. Therefore, the resulting plasmons are physically similar to two coupled oscillators. The obtained plasmon branches are defined as acoustic (lower frequencies) and optical (higher frequencies), or as in-phase and out-of-phase [65]. The number of branches is equal to the number of layers [66], or, more precisely, the number of separate plasmon excitations in them. However, one cannot attribute one branch to the first layer and the other to the second one.

We should also mention that the frequency of the optical plasmon branch depends on the distance between the layers. One can easily verify this by analyzing how far the branch moves from the main diagonal $\omega =v_{F}q$ when the distance between the layers is increased. However, this dependence is not as dramatic as for the acoustic plasmons, which also change their shape when the distance between the layers is increased, and the Coulomb coupling is faded away. Finally, we should say that the main subject of the present paper is the plasma branches for a system of different layers and their damping rates, which are affected by the distance between the layers and the type of substrate in between.

To extract more information about the plasmonic behavior, in Figure 8a, we have presented the figure to show the result highlighting the changes in the plasmonic nature for a structure with different types of layers and substrates. In Figure 8a, we demonstrate the plasmon mode for a structure with silicene and graphene separated by a distance of $1.0{\left(k_{F}^{\left(0\right)}\right)}^{-1}$ with the vacuum in between. One can observe two plasmon modes originating from the origin in q−ω space.

A special effect of overcoming the single-particle excitation region of silicene by the single-particle excitation region of graphene is observed, which causes the shortening of the plasmon branches that used to be there for the silicene–silicene structure. As soon as the plasmon branch reaches the particle-hole mode region, the plasmon mode decays by replacing one silicene layer with a graphene layer in the silicene–silicene structure. The effect, due to the band gap in silicene, is just nullified. In other words, the plasmon modes in the regime, where they used to have plasmon modes before, now do not have them, because the plasmon mode decays into the particle-hole mode of graphene. Furthermore, the result of adding a substrate between the silicene and graphene layer is illustrated in panel (b) of Figure 8. In this case, we could see a new plasmon branch originating from the bulk plasma frequency and one plasmon branch originating from the origin. Here, due to the presence of a substrate, the lower plasmon branch bends sharply toward the intraband single-particle excitation region where it decays causing complete disappearance. The upper plasmon branch and the plasmon from the bulk plasmon frequency become closer and move toward the interband single-particle excitation region where they become damped.

The results in Figure 8 correspond to a graphene layer located on the top and the silicene layer located below it in the heterostructure. The order in which the layers are placed affects the plasma dispersions only if asymmetric substrates are involved. In contrast, the plasmon dispersion relation for a pair of Coulomb-coupled layers embedded in a uniform background is determined by a 2×2 determinant equation [67], which is symmetric to switching the layers. This is what our Equation (Equation 9) is reduced to when $\epsilon_{1}=\epsilon_{2}=1$ and the background dielectric constants are independent of the frequency ω.

These new effects on the plasmon branches in this type of structure were not reported previously. Results of this type help develop electronic and quantum computing devices where knowledge of the plasmonic behavior of materials and their damping nature is very essential.

## 4. Concluding Remarks

In summary, we have investigated the key properties of the plasmonic mode and damping for different combinations of graphene and silicene layers. The effect of the addition of substrate in between the two layers is further analyzed. This resulted in the development of a novel technique of tuning the plasmon excitation mode associated with the two dimensional systems. In our system, a complete new plasmon branch emerges from the bulk plasmon frequency, this would be very helpful in engineering modern computing devices.

Another discovery we have from this study is the disappearance of the lower plasmon branch and the suppression of the silicene band gap effect. Along with the study of interesting features in the plasmon modes, we have also developed an approach for calculating the decay rates for the plasmons due to Landau damping by the particle-hole modes.

The principal goal of our investigation and the main results of our work are concerned with the understanding of plasmonic nature and analyzing their Landau damping rates in a multi-layer structure. In contrast to Ref. [67], which includes only the long-wavelength limit for graphene without any discussion of gapped materials (such as silicene), our work is concerned with a thorough and detailed investigation of plasmon and damping rates for finite-value wave vectors and energies.

Additionally, our results infer that the number of plasmon branches emerging from the origin can be varied by choosing the number of the 2D layer. In brief, we can say that our study gives an important idea about the plasmonic behavior of a graphene–silicene-based heterostructure, which would be very helpful in carrying out further studies of other types of heterostructure, including a variety of low dimensional materials.

## Figures and Tables

**Figure 1 materials-15-07964-f001:**
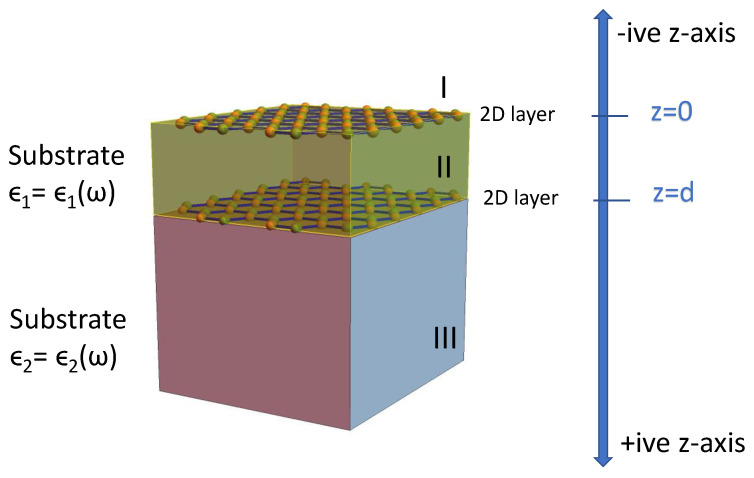
(Color online) Schematic illustration of a heterostructure consisting of a pair of 2D layers separated by a dielectric medium $\epsilon_{1}\left(\omega \right)$. This structure lies on a substrate with a dielectric function $\epsilon_{2}\left(\omega \right)$.

**Figure 2 materials-15-07964-f002:**
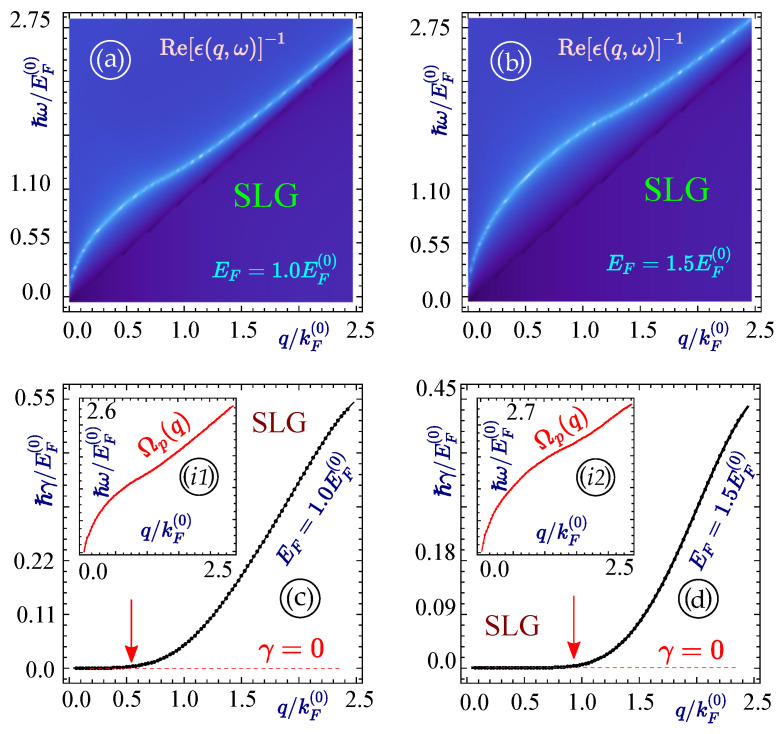
(Color online) Plasmon frequency $\omega_{p}\left(q\right)$ and damping rates $\gamma (\omega_{p}\left(q\right),q)$ for an isolated graphene layer (SLG) with $E_{F}=1.0E_{F}^{\left(0\right)}$(left panels (**a**,**c**)) and $E_{F}=1.5E_{F}^{\left(0\right)}$(right panels (**b**,**d**)). Two top panels (**a**,**b**) demonstrate the plasmon dispersion (either damped or undamped obtained as Re(ϵ(q,ω))=0, the lower plots (**c**,**d**) describe the corresponding damping rate along the plasmon branches, also calculated and shown as insets (i1) and (i2).

**Figure 3 materials-15-07964-f003:**
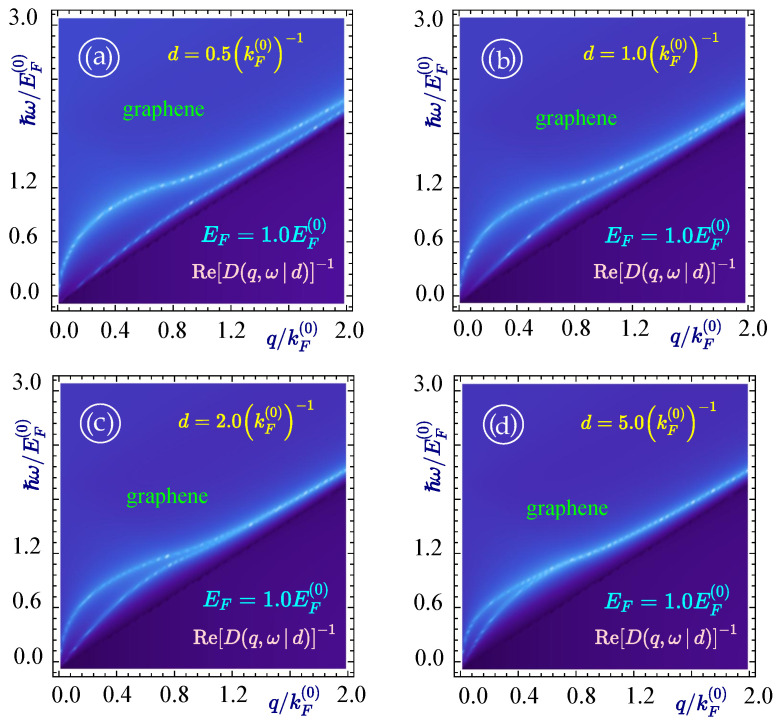
(Color online) Acoustic (**lower**) and optical (**upper**) plasmon modes for a pair of identical graphene layers with $E_{F}=1.0E_{F}^{\left(0\right)}$. Each panel corresponds to different values of the separation between the layers corresponding to (**a**) $d=0.5{\left(k_{F}^{\left(0\right)}\right)}^{-1}$, (**b**) $1.0{\left(k_{F}^{\left(0\right)}\right)}^{-1}$, (**c**) $2.0{\left(k_{F}^{\left(0\right)}\right)}^{-1}$ and (**d**) $5.0{\left(k_{F}^{\left(0\right)}\right)}^{-1}$, as labeled. The plasmon dispersion (either damped or undamped) is obtained by solving Re(D(q,ω|d))=0.

**Figure 4 materials-15-07964-f004:**
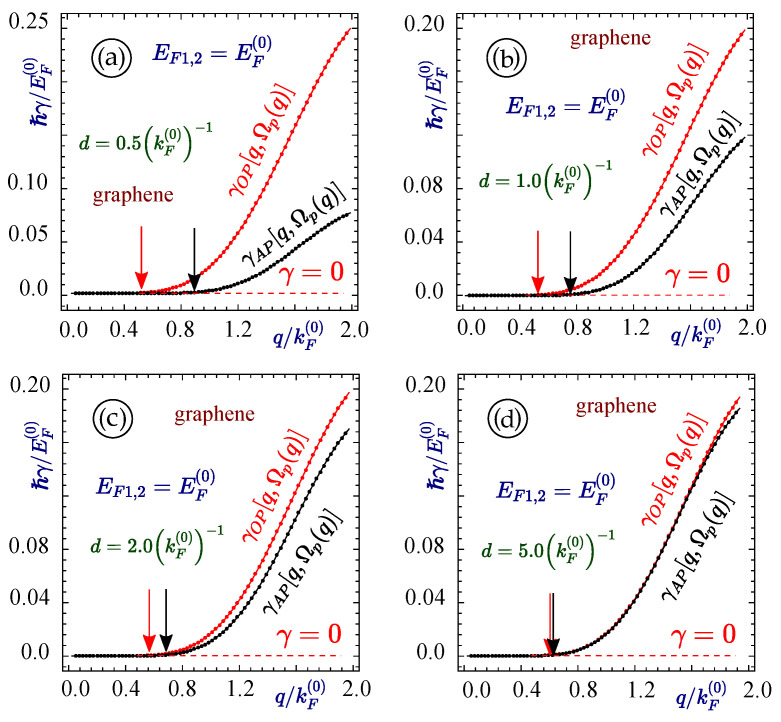
(Color online) The damping rates corresponding to the acoustic and optical plasmon branches shown in Figure 4 for two identical graphene layers with $E_{F}=1.0E_{F}^{\left(0\right)}$. Each panel corresponds to a different values of the separation between the layers with (**a**) $d=0.5{\left(k_{F}^{\left(0\right)}\right)}^{-1}$, (**b**) $1.0{\left(k_{F}^{\left(0\right)}\right)}^{-1}$, (**c**) $2.0{\left(k_{F}^{\left(0\right)}\right)}^{-1}$ and (**d**) $5.0{\left(k_{F}^{\left(0\right)}\right)}^{-1}$, as labeled.

**Figure 5 materials-15-07964-f005:**
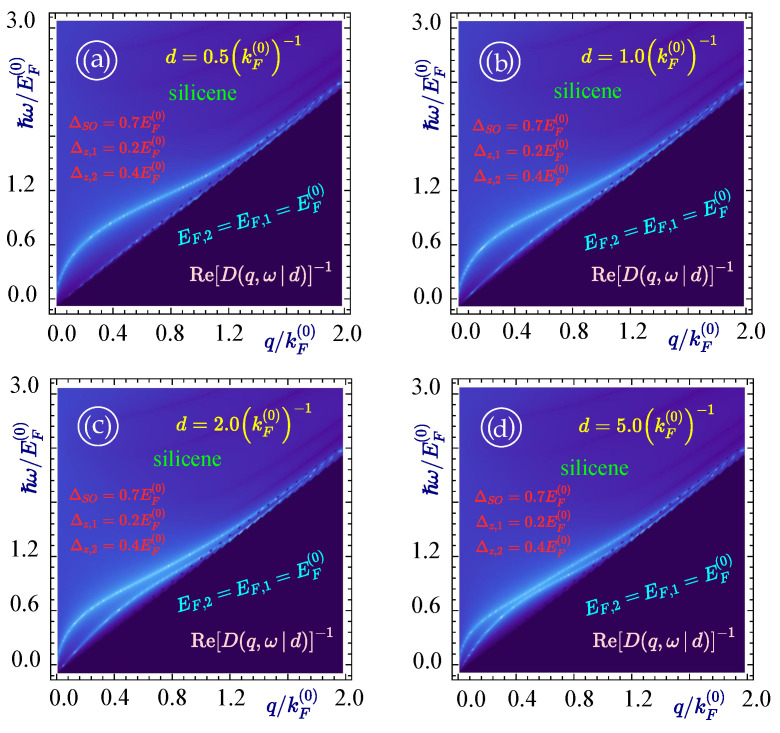
(Color online) Acoustic (**lower**) and optical (**upper**) plasmon dispersions for two silicene layers with $E_{F}=1.0E_{F}^{\left(0\right)}$ and the band gaps $\Delta_{SO,1}=\Delta_{SO,2}=0.7E_{F}^{\left(0\right)}$, $\Delta_{z,1}=0.2E_{F}^{\left(0\right)}$ and $\Delta_{z,2}=0.4E_{F}^{\left(0\right)}$. Each panel corresponds to different values of the separation between the layers (**a**) $d=0.5{\left(k_{F}^{\left(0\right)}\right)}^{-1}$, (**b**) $1.0{\left(k_{F}^{\left(0\right)}\right)}^{-1}$, (**c**) $2.0{\left(k_{F}^{\left(0\right)}\right)}^{-1}$ and (**d**) $5.0{\left(k_{F}^{\left(0\right)}\right)}^{-1}$, as labeled.

**Figure 6 materials-15-07964-f006:**
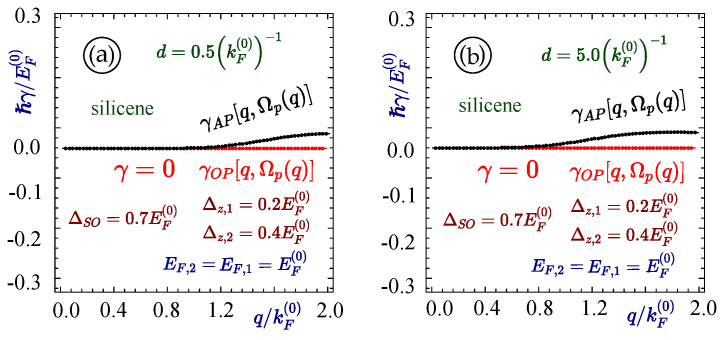
(Color online) The damping rates corresponding to the acoustic and optical plasmon branches calculated in Figure 6 for two silicene layers with $E_{F}=1.0E_{F}^{\left(0\right)}$, and the band gaps $\Delta_{SO,1}=\Delta_{SO,2}=0.7E_{F}^{\left(0\right)}$, $\Delta_{z,1}=0.2E_{F}^{\left(0\right)}$ and $\Delta_{z,2}=0.4E_{F}^{\left(0\right)}$. Each panel corresponds to different values of the separation between layers with (**a**) $d=0.5{\left(k_{F}^{\left(0\right)}\right)}^{-1}$ and (**b**) $5.0{\left(k_{F}^{\left(0\right)}\right)}^{-1}$, as labeled.

**Figure 7 materials-15-07964-f007:**
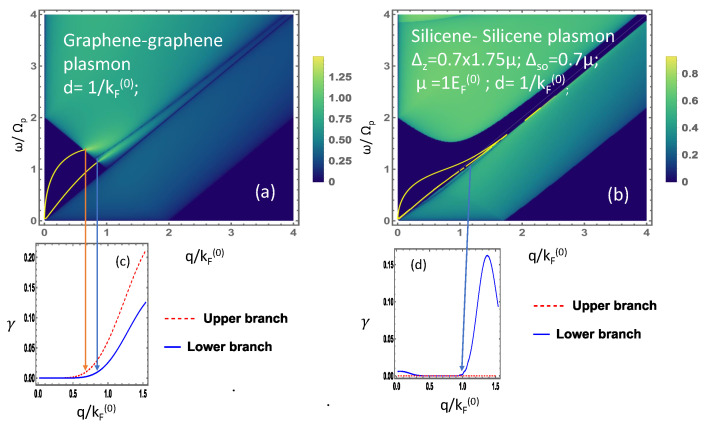
(Color online) Plasmon modes for (**a**) graphene–graphene structure and (**b**) silicene–silicene structure with corresponding plasmon damping rate in (**c**) and (**d**), respectively.

**Figure 8 materials-15-07964-f008:**
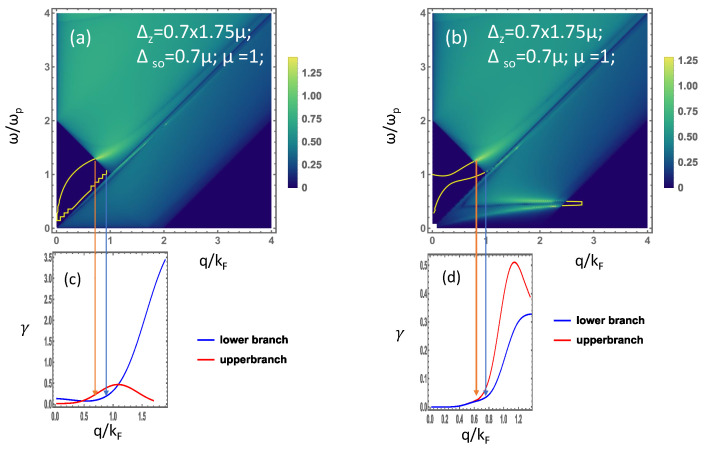
(Color online) Plasmon mode dispersion for a heterostructure. (**a**) Graphene-vacuum-silicene, (**b**) graphene-substrate-silicene with the dielectric function, $\epsilon_{1}\left(\omega \right)=1-\Omega_{p}^{2}/\omega^{2}$, for the substrate. Panels (**c**,**d**) represent the plasmon damping rate for the structure corresponding to the plasmon modes in (**a**) and (**b**), respectively. The bottom substrate is given by $\epsilon_{2}=1$ for both cases.

## Data Availability

Exclude this statement as the study did not report any data.

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
