# Peer review of "Plasmon Damping Rates in Coulomb-Coupled 2D Layers in a Heterostructure"

_materials, 2022, doi:10.3390/ma15227964_

Round 1

Reviewer 1 Report

Comments to author (if any)

This manuscript presents “Plasmon damping rates in Coulomb coupled 2D layers in a heterostructure”. The formulated products were characterized via several techniques and the findings are well explained and supported to results, language used is good. The following comments should be considered before its publishment in Materials with Minor revision

1- It would be nice if authors provide some novelty and motivation lines in abstract

2. The language used is good

3. Images are very good quality with comprehensive figure caption. All are linked to the text but please increase the inside writing size of figure 7.

4. The references are good but to increase the worth of manuscript, incorporate the provided articles in introduction section

(a) 10.1007/s13204-019-01239-3 (b) 10.1016/j.jallcom.2020.155588

Author Response

Dear Reviewer,

Thank you for your time and advice to improve the value of our paper. We appreciate it!! Here are our responses to your comments point by point.

Your comments:

This manuscript presents “Plasmon damping rates in Coulomb coupled 2D layers in a heterostructure”. The formulated products were characterized via several techniques and the findings are well explained and supported to results, language used is good. The following comments should be considered before its publishment in Materials with Minor revision

1- It would be nice if authors provide some novelty and motivation lines in abstract.

Our response:  Following your suggestions, we have added a few lines in regard to the novelty and the motivation in the abstract. The following lines are added:

" A novel effect of nullification of the silicene band gap is noticed
when graphene is introduced into the system. To utilize these effects for experimental and industrial
purposes, graphical results for the different parameters are presented".

  1. The language used is good.                                                                             Our response:  Thank you for your encouraging comments about the paper.
  2. Images are very good quality with comprehensive figure caption. All are linked to the text but please increase the inside writing size of figure 7.        Our response: Following your suggestions, we have increased the font size of the text inside the figure from 18 to font size 24. You can see that in the revised version.
  3. The references are good but to increase the worth of manuscript, incorporate the provided articles in introduction section

(a) 10.1007/s13204-019-01239-3 (b) 10.1016/j.jallcom.2020.155588

 Our response:  Thank you for your suggestion. We have added the mentioned references in the manuscript accordingly which you can see in reference 40 and 41 of the reference list.

Thank you

sincerely

Dipendra Dahal

corresponding author

Reviewer 2 Report

In this work, authors demonstrate the way in which the Coulomb interaction renormalizes the plasmon frequencies and investigated how the decay rates of the plasmons in these heterostructures are affected by the Coulomb coupling between different types of two-dimensional materials. the calculation are clear, which can surpport  the conclusions of the Coulomb excitations of charge density oscillation  between two-dimensional (2D) layers of silicene and graphene. I think this work should be accepted in present form.

Author Response

Dear reviewer, 

Thank you for your time and effort to review our paper and for making encouraging comments about our work. We greatly appreciate your time in going into detail about our research work.

thank you

sincerely

Dipendra Dahal

corresponding author

Reviewer 3 Report

The authors have calculated the plasmon dispersion relations and damping rates of graphene, silicene and their (hetero-bilayers), in vacuum and on substrate using model Hamiltonians. According to the authors, the method of calculating the damping rate as well as the corresponding results for the multilayers are novel (but I am not sufficiently familiar with the literature on graphene plasmon calculations to be able to comment further). In addition, the authors find curious interplay between the plasmon modes in the "heterostructures" of (non-gapped) graphene and (gapped) silicene.

The manuscript is reasonably well-written with only occasional typos and other minor errors, and the results clearly presented and analyzed. I did not spot any problems in the theory or computational methods.

The reported results may not be directly relevant to experimentalists as they depend on many parameters related to experimental details, the results (and methods) give useful insight to the damping in multilayer systems. Thus, I think the manuscript is suitable for publication in Materials.

Below are few minor detailed comments:

- page 2: "Recently, Dong et. al. has studied the plasmonic and its decay mode...", plasmonic what?

- page 2: I could not find the adopted numerical values for \Delta_so and \Delta_z.

- It might be useful to add color bars to Figs. 2, 3, and 5.

- Fig. 3 (and discussion) Please indicate d also in the units of [Å] or [nm] to give an immediate impression of the separation (without reader having to calculate it).

- page 9: "plasmode mode"

Author Response

Dear reviewer,

Thank you for your time and effort in going into detail about our research work. We really appreciate your valuable suggestions in making our paper more worthful. Following your comments, we have made the changes accordingly in the manuscript. 

Here is how we address your comments:

Your comments:

The authors have calculated the plasmon dispersion relations and damping rates of graphene, silicene and their (hetero-bilayers), in vacuum and on substrate using model Hamiltonians. According to the authors, the method of calculating the damping rate as well as the corresponding results for the multilayers are novel (but I am not sufficiently familiar with the literature on graphene plasmon calculations to be able to comment further). In addition, the authors find curious interplay between the plasmon modes in the "heterostructures" of (non-gapped) graphene and (gapped) silicene.

The manuscript is reasonably well-written with only occasional typos and other minor errors, and the results clearly presented and analyzed. I did not spot any problems in the theory or computational methods.

The reported results may not be directly relevant to experimentalists as they depend on many parameters related to experimental details, the results (and methods) give useful insight to the damping in multilayer systems. Thus, I think the manuscript is suitable for publication in Materials.

Below are few minor detailed comments:

  • page 2: "Recently, Dong et. al. has studied the plasmonic and its decay mode...", plasmonic what?                                                                               Our response: Thank you for showing us the typo in the manuscript, we have corrected it in the revised version.

- page 2: I could not find the adopted numerical values for \Delta_so and \Delta_z.                                                                                                                      Our response:   In the current version we have presented the range of numerical values for the band gap \Delta_so with the reference for it. The value of \Delta_z is related to the value of \Delta_so which are illustrated in the manuscript and in the figures as well.

  • It might be useful to add color bars to Figs. 2, 3, and 5.                                 Our response:  We thank you for looking in detail about the paper and the figures. Here in this case figs. 2-5 demonstrate a plasmon branch that is partially (or sometimes even completely) undamped. Ideally, the presented D function jumps to infinity at the points of a plasmon branch.  The real value of the plotted function in a realistic graph is high but finite, its actual values are only determined by the precision of our calculation. Therefore, the actual numerical values for our density plots and the color bars are meaningless.  That is why we have only shown the location of the two plasmon branches in the interacting 2D layers. 
  • Fig. 3 (and discussion) Please indicate d also in the units of [Å] or [nm] to give an immediate impression of the separation (without the reader having to calculate it).                                                                                                  Our response: Thank you for your suggestions to make this paper easily and understandable for all readers.  Appreciating  your comments we now have presented the real value of the thickness of the interlayer separation in the text. 
  • page 9: "plasmode mode"                                                                                 Our response:  We have corrected the mentioned typo and further proofread to determine if there are any other typos.                                        With thanks                                                                                                     sincerely                                                                                                             Dipendra Dahal                                                                                              corresponding author.
